# Neural Routing by Memory

**Kaipeng Zhang**[1,2]    **Zhenqiang Li**[1]    **Zhifeng Li**[2*]    **Wei Liu**[2*]    **Yoichi Sato**[1*]

[1]Institute of Industrial Science, The University of Tokyo
[2]Tencent Data Platform

{kpzhang, lzq}@iis.u-tokyo.ac.jp    michaelzfli@tencent.com
wl2223@columbia.edu    ysato@iis.u-tokyo.ac.jp

## Abstract

Recent Convolutional Neural Networks (CNNs) have achieved significant success by stacking multiple convolutional blocks, named procedures in this paper, to extract semantic features. However, they use the same procedure sequence for all inputs, regardless of the intermediate features. This paper proffers a simple yet effective idea of constructing parallel procedures and assigning similar intermediate features to the same specialized procedures in a divide-and-conquer fashion. It relieves each procedure's learning difficulty and thus leads to superior performance. Specifically, we propose a routing-by-memory mechanism for existing CNN architectures. In each stage of the network, we introduce parallel Procedural Units (PUs). A PU consists of a memory head and a procedure. The memory head maintains a summary of a type of features. For an intermediate feature, we search its closest memory and forward it to the corresponding procedure in both training and testing. In this way, different procedures are tailored to different features and therefore tackle them better. Networks with the proposed mechanism can be trained efficiently using a four-step training strategy. Experimental results show that our method improves VGGNet, ResNet, and EfficientNet's accuracies on Tiny ImageNet, ImageNet, and CIFAR-100 benchmarks with a negligible extra computational cost.

## 1  Introduction

Human memory is often understood as an informational processing system. It plays an essential role in human intelligence and comprises short-term memory and long-term memory, inspiring many well-known machine learning models, such as Recurrent Neural Networks (RNN), Long Short-Term Memory (LSTM) [14], and Neural Turing Machine (NTM) [11]. Episodic memories, a type of long-term memory, are the collection of past personal experiences. They can be retrieved and exploited by the brain when tackling problems that have been encountered before. Different memories activate different neurons in the brain, directing us to perform specific procedures that we have done before. Inspired by this observation, we introduce the routing-by-memory mechanism to the neural network. It uses memory (a summary of seen features) to guide networks to process different features with different procedures in a divide-and-conquer manner, easing the difficulty of learning and achieving better performance. This paper applies the mechanism to the feature extraction in CNNs and refers to networks employing it as Routing-by-Memory Networks (RMNs). In the following, we will introduce conventional CNNs and our RMN.

Recent Convolutional Neural Networks (CNNs) achieve state-of-the-art results on computer vision tasks by stacking convolutional blocks (e.g., residual block [13], inception block [33], and Dense block [17]), named procedures in this paper, to extract semantic features. However, they use the same

---

*    Co-corresponding authors

35th Conference on Neural Information Processing Systems (NeurIPS 2021).

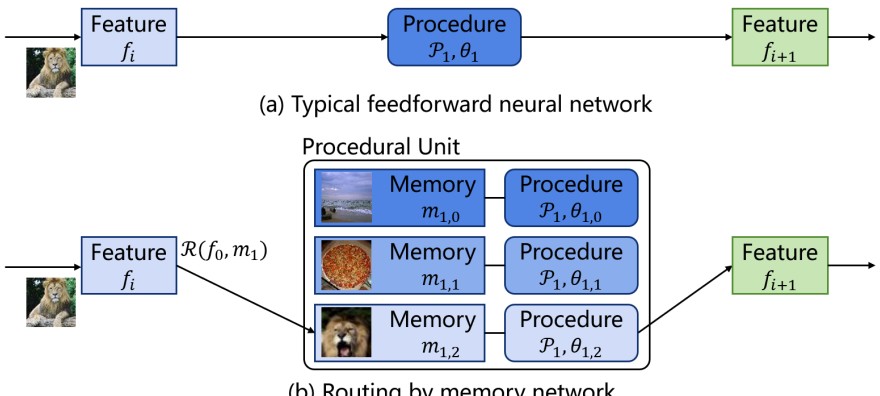

(a) Typical feedforward neural network

Procedural Unit

(b) Routing by memory network

Figure 1: (a) Conventional feed-forward network. It processes the features by stacked procedures. (b) Our proposed Routing by Memory Network (RMN). It employs parallel procedures (i.e., multi-branch network) and processes features in a divide-and-conquer manner. Different procedures are tailored to different types of features. $\mathcal{R}$ is a routing function, and we use the nearest neighbor algorithm for it. $m$, $\mathcal{P}$, and $\theta$ denote the memory, procedure architecture, and the parameters of the procedure, respectively. Memory indicates which type of features the corresponding procedure can handle. Given a feature that a previous stage yields, it searches its most similar memory and forwards the feature to the corresponding procedure. For example, $m_{1,0}$, $m_{1,1}$, and $m_{1,2}$ represent scenery, food, and animal, respectively. Given a lion's features, $\mathcal{R}$ will forward them to the third procedure.

stacked blocks to process all intermediate features, despite large variances of those features. Since similar intermediate features can be processed in the same way, we propose a simple yet effective idea of introducing parallel procedures (i.e., multi-branch CNN) and assigning similar intermediate features to the same specialized procedures (i.e., the same branch). It is a divide-and-conquer structure, seeing Figure 1 for an illustration. In this way, we can improve the model capacity and performance while not increasing the computational cost since we forward an intermediate feature to only one procedure.

In our RMN, we introduce a simple yet effective mechanism, routing by memory. Briefly, we introduce the Procedural Unit (PU) to process features (see Figure 1 and Figure 2 for illustrations). It consists of a memory (a summary/representative feature) with a procedure (some convolutional blocks). We use the memory to identify which type of features the corresponding procedure is expected to handle. Specifically, we split the network into different stages by downsampling layers. There are multiple PUs in each stage. All procedures within a stage use the same architecture but different parameters. In a stage, given an intermediate feature produced from the previous stage, we will search its nearest memory and forward it to the corresponding procedure. In this way, different PUs are specialized to handle different types of features. Besides, in the procedures, we introduce a routing-dependent Squeeze-and-Excite (SE)-like feature attention module, dubbed Conditional Attention (CA), to improve the performance further.

How to initialize and update memory and procedures is the main challenge when training our RMN. We propose an easy-to-implement training strategy that includes four training steps: stem network training, procedure cloning, memory initialization, and routing-based training. In the stem network training step, we train a conventional CNN (e.g., ResNet and EfficentNet) as a stem network. Then in the procedure cloning step, we generate multiple procedures (i.e., multi-branch CNN) in each stage of the network by cloning all learned procedures a preset number of times. In the memory initialization step, we first extract intermediate features for all stages of the network. Then we use representative features as initialized memory. This paper clusters features in each stage of the network and uses cluster centers as representative features. Finally, we continue the network training in the routing-based training step, which is similar to the first step, but the data flow inside the network is decided by routing results. In this step, the memory is updated in a moving average fashion while other components are updated by the original optimization method used in the first stage. The overall training strategy is plug-and-play to existing CNNs' training strategies.

This paper takes VGGNet, ResNet, and EfficientNet as backbones to train our proposed RMN. According to the experimental results, RMN significantly improves original models' results on some benchmarks while not increasing the computational cost. We summarize our contributions as follows:

- We propose a novel divide-and-conquer mechanism called routing by memory, which constructs a multi-branch CNN and assigns similar features to the same specialized branch. It can lead to better performance while not increasing the computational cost.
- The proposed mechanism is plug-and-play to existing CNN architectures by virtue of a proposed effective four-step training strategy.
- We apply our mechanism to VGGNet, ResNet, and EfficientNet and achieve significant improvements in the accuracies on Tiny ImageNet, Imagenet, and CIFAR-100 benchmarks.

## 2 Related Work

### 2.1 Convolutional Neural Network Architecture

In 2012, AlexNet [23] outperforms handcrafted features engineering by a large margin in the ImageNet image classification competition. It combines feature extraction and classification in a deep CNN model to extract discriminative semantic features. After that, CNNs gradually dominate most computer vision tasks, such as image classification, object detection, and semantic segmentation. Classic CNN architecture (e.g., AlexNet [23], ZFNet [41], and VGGNet [32]) extract features by stacking convolutional layers. In 2016, ResNet [13], the most successful CNN architecture in recent years, proposed to train very deep CNNs using skip connections and stacked residual blocks. After that, more and more efficient and effective stacked block-based CNNs (e.g., MobileNet[15], SENet [16], and EfficientNet [34]) were proposed. However, they use the same block sequence for all inputs in feature extraction, regardless of the intermediate features. This paper introduces the plug-and-play routing-by-memory mechanism to existing CNN architectures. It uses memory to guide different blocks to tackle different features in a divide-and-conquer fashion, which boosts the accuracy while not increasing the computational cost.

### 2.2 Conditional Computation

Conditional computation [3] refers to activating only some of the modules in a network in an input-dependent fashion. Recent researches have introduced it to CNNs in order to accelerate network inference. AIG [36], BlockDrop [39], and SkipNet [37] propose to learn the subset of blocks needed to process a given input. Since easy examples may not require deep layers' features to make the classification, SACT [9], Inside Cascaded CNN, [42] and Dynamic Routing [26] propose to do input-dependent early stopping at the stage of network inference. Routing Convolutional Network (RCN) [20] introduces a routing network that can perform anytime recognition. These methods aim to reduce the models' computational costs while maintaining comparable accuracies.

### 2.3 Mixture of Experts

Mixture of Experts (MoE) ([19], [21]) was proposed three decades ago. It is related to conditional computation and introduces multiple experts (learners) to divide the problem space into homogeneous regions. Given an input, MoE often uses a routing module to select the corresponding experts in a hard or soft selection fashion. In the deep learning era, sparsely-gated mixture-of-experts layer [31] first introduces MoE to LSTM for language modeling. Lately, Switch Transformer [8] introduces MoE to Transformer [35] to train a trillion parameter model and achieves great success in natural language processing. As for computer vision, there are two categories of MOE, dynamic parameter and dynamic architecture, which are compatible. Dynamic parameter methods [40, 5, 6] learn input-dependent convolutional kernels' parameters using parameters generation networks. Dynamic architecture methods employ multiple sub-networks and build a dynamic computational graph by decision networks. NOEF [1] and HydraNets [27] divide the class label space and assign different sets of class labels to different sub-networks for handling. DeepMoE [38] proposes to increase the kernels and select each layer's features' channels by a decision network. Runtime Routing [28] proposes to learn an RNN-based decision network by reinforcement learning for multi-path networks. Akin to the Runtime Routing, DRNet [4] uses a small CNN as the decision network. Dynamic Routing [25],

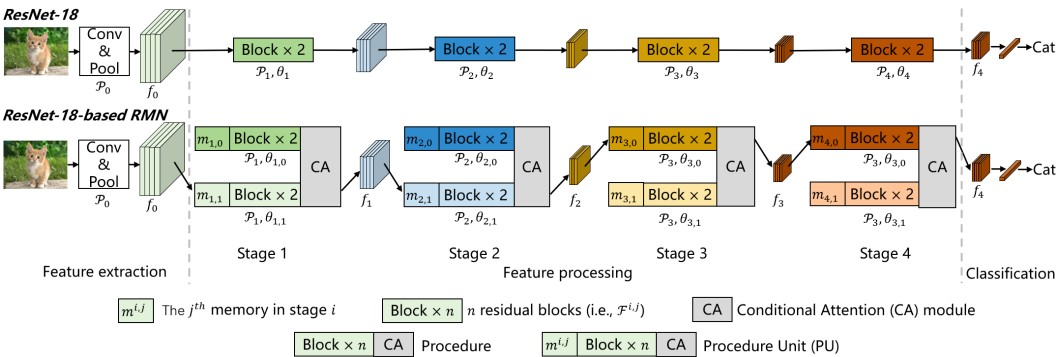

Figure 2: An illustration of ResNet-18 and ResNet-18-based 2-way RMN. We divide the network into three phases: (preliminary) feature extraction, feature processing, and classification. In the feature processing phase, we split the network into four stages by downsampling layers. RMN introduces the routing-by-memory mechanism to each stage. It uses memory to guide forwarding different features to their corresponding specialized procedures. Different colors denote different stages. Different shades of color represent different kinds of features, memories, and procedures within a stage.

Routing Networks [29], ExperGate [2], and HardMoE [12] extend MoE to semantic segmentation, multi-task learning, weakly-supervised learning, and lifelong learning, respectively. These decision networks increase the training difficulty and bring extra computations. Besides, some of them require modifying relevant gradient descent algorithms or employing reinforcement learning to train the decision networks.

Our RMN solves routing in a new perspective, routing by memory, rather than designing complicated decision networks to decide the routing path. It is more straightforward and elegant. Besides, since the routing is based on non-parametric nearest memory search, RMN is lightweight, easy to train, and requires no modification in the back-propagation rule.

## 3   Routing by Memory Network

This section elaborates on the architecture and training strategy of our proposed RMN.

### 3.1   Network Architecture

Our RMN is built by integrating the routing-by-memory mechanism into an existing CNN architecture. In this paper, we describe our methods with the task of image classification. According to the functions, we treat the processing of a common CNN as three separated phases: (preliminary) feature extraction, feature processing, and final classification. Our RMN enhances the intermediate features in the feature processing phase by incorporating the routing-by-memory mechanism. In the following sections, we take ResNet-18 [13] as an example backbone to present how to integrate the routing-by-memory mechanism into a conventional CNN(see Figure 2).

#### 3.1.1   ResNet

In this section, we elaborate on the ResNet-18. Given an input image $\mathcal{X}$, in the first (preliminary) feature extraction phase, its feature maps $f_0$ are generated by a series of operations $\mathcal{P}_0$ consisting of convolution, pooling, and batch normalization, which can be formulated as follows:

$$f_0 = \mathcal{P}_0(\mathcal{X}, \theta_0), \tag{1}$$

where $\theta_0$ denotes the learnable parameters in $\mathcal{P}$. Low-level features such as edges and corners are represented in these preliminary feature maps.

In the feature processing phase, multiple residual blocks are utilized to process the feature map $f_0$ for extracting features in higher semantic levels. We group the blocks into different stages according to their output feature maps' dimensions and resolutions (i.e., divide the stages by down-sampling operations). For example, as shown in Figure 2, we divide blocks into four stages by grouping

two adjacent blocks as their feature maps' shapes are the same. We represent the operations (i.e., convolution blocks) in the $i^{th}$ stage as $\mathcal{P}_i$ with its parameters $\theta_i$. The feature maps yielded from the $i^{th}$ stage are $f_i$. Specifically, we formulate the processing of the $i^{th}$ stage as follows:

$$f_i = \mathcal{P}_i(f_{i-1}, \theta_i), \qquad i = 1, 2, 3, 4. \tag{2}$$

The ResNet ends with a global average pooling layer and a 1000-way (the number of classes, 1000 in ImageNet) fully-connected layer with an argmax (softmax in the training phase) function. They are used to classify one given image in the classification phase. We formulate this phase as follows:

$$\mathcal{Y} = \underset{c \in [0,1,...,C-1]}{\arg\max} \ (\mathrm{GAP}(f_4) W_c^\top + b), \tag{3}$$

where GAP refers to the global average pooling in the classification phase. $W$ and $b$ are learnable weights and biases of the fully-connected layer (i.e., classifier), respectively. $c$ is the class index, and $C$ is the total number of classes. $\mathcal{Y}$ denotes the output class label.

### 3.1.2  ResNet-Based RMN

In this section, we take the ResNet-18 as a backbone network to introduce our RMN. In the Resnet-18-based RMN, we propose Procedural Units (PU) in the feature processing phase. There are multiple PUs in each stage, and we use the term $N$-way RMN to represent how many PUs per stage in our RMN. All PUs within a stage share the same architecture but learn different parameters due to the routing mechanism. Each PU consists of two modules, including memory and procedure. The memory is a representative feature that is learned from features by a moving average fashion in the training phase (see Section 3.2 for details). We use global average pooling for memory to reduce the storage cost. In the $i^{th}$ stage, given the feature $f_{i-1}$ and memories $m_i = [m_{i,0}, m_{i,1}, ..., m_{i,N-1}]$, we first do the routing $\mathcal{R}$ by nearest neighbor searching as follows:

$$\mathcal{R}(m_i, f_{i-1}) = \underset{j \in [0,1,...,N-1]}{\arg\min} \ (\mathcal{D}(m_{i,j}, \mathrm{GAP}(f_{i-1}))), \tag{4}$$

where $m_{i,j}$ denotes the $j^{th}$ PU's memory in the $i^{th}$ stage. To reduce the storage cost, we apply global average pooling GAP to $f_{i-1}$ before routing. $\mathcal{D}$ refers to a distance measurement metric, and we use Euclidean distance in this paper. Since $\mathcal{D}$ is non-parametric and fast to compute, the computational cost of routing is negligible and can be omitted.

After the routing, we forward the features to the corresponding procedures. A procedure consists of residual blocks and an optional Conditional Attention (CA) module. The CA module is an optional module introduced to improve the accuracy further by a little additional computational cost. Without the CA module, RMN re-formulates the $i^{th}$ stage of the feature processing phase as follows:

$$f_i = \mathcal{P}_i(f_{i-1}, \theta_{i,\mathcal{R}(m_i, f_{i-1})}), \qquad i = 1, 2, 3, 4, \tag{5}$$

where $\theta_{i,0}, \theta_{i,1}, ..., \theta_{i,N-1}$ refers to $N$ sets of parameters for $\mathcal{P}_i$.

We then introduce the CA module. The same channels of features produced from different procedural units may represent different semantic meanings. For example, the first convolution kernel of a procedure may focus on animals' fur. However, the first convolution kernel of another procedure may focus on furniture's texture. The inconsistent semantic meaning of different features increases the learning difficultly. Inspired by the position-coding in ViT [7], we introduce the CA module to do routing-dependent channel-wise attention to the features, relieving the inconsistency adaptively. The CA module consists of a conditional input, a routing result, and a Squeeze-and-Excitation (SE) module [16]

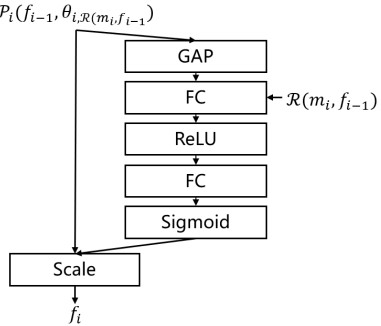

Figure 3: An illustration of Conditional Attention (CA) module. The CA is a routing-dependent Squeeze-and-Excitation (SE)-like module.

(see Figure 3 for an illustration of the CA module). Specifically, it first computes the scalars $s = [s_1, s_2, ..., s_K]$ that has the same number of channels as $f_i = [f_{i_1}, f_{i_2}, ..., f_{i_K}]$. This operation can be formulated as:

$$s = Excite(Concat(Squeeze(f_i), \mathcal{R}(m_i, f_{i-1}))), \tag{6}$$

where $Squeeze$ denotes using a global average pooling followed with a fully-connected layer to squeeze the feature dimension. $Excite$ denotes using a fully-connected layer and a Sigmoid function to expand the feature dimension and compute attention values. $Concat$ denotes channel-wise concatenation operation. It can also use different coding methods for the routing results, such as one-hot coding, which sometimes makes the training more stable. Please note that all CA modules share parameters within a stage.

Then we scale the features $f_i$ according to the scalar vector $s$ by channel-wise multiplication (we use $*$ to represent it). The CA module can be formulated as follows:

$$CA(f_i) = s * f_i. \tag{7}$$

In summary, when using the CA module, RMN re-formulates the $i^{th}$ stage of the feature processing phase as follows:

$$f_i = s * \mathcal{P}_i(f_{i-1}, \theta_{i,\mathcal{R}(m_i, f_{i-1})}), \quad i = 1, 2, 3, 4. \tag{8}$$

## 3.2 Training Strategy

In this section, we take the ResNet-18-based RMN as an example to introduce the four-step training strategy. Unlike the conventional training pipeline, after a few training epochs, we insert two steps to expand procedures and initialize memories. In the last step, we update memories in a moving average fashion. The training strategy is plug-and-play to existing CNNs' training approach since it can follow standard gradient descent algorithms in optimization.

### 3.2.1 Stem Network Training

In the first step, we aim to train a conventional CNN (e.g., VGG, ResNet, and EfficientNet), named stem network in this paper, to extract reasonable features for memory initialization. Specifically, in this step, we train the standard ResNet-18 for a few epochs and will resume the training in the fourth step. If using CA modules, we use random routing results $r \sim \mathcal{U}(1, N)$ for each CA module, where $\mathcal{U}$ denotes the discrete uniform distribution.

### 3.2.2 Procedure Cloning

We clone the procedures in the second step. Specifically, we clone the parameters of all procedures by $N$ times (i.e., assign $\theta_i$ to $\theta_{i,0}, \theta_{i,1}, ..., \theta_{i,N-1}$). In this way, we can build $N$ branch procedural units in each stage of the feature processing phase. Please note that we also clone the optimizer state of the gradient in this step to maintain the training's consistency in the fourth phase.

### 3.2.3 Memory Initialization

We initialize the memories using representative features. In this paper, we compute the representative features by cluster analysis. Specifically, in the $i^{th}$ stage of the feature processing phase, we extract each training sample's $f_i$. Then we apply the Euclidean distance-based K-means clustering algorithm to obtain $N$ clusters. Finally, we initialize each memory in $m_{i+1}$ by each cluster' center. In this way, different memories can dominate different kinds of features.

### 3.2.4 Routing-Based Training

In this step, we resume the model training. Inspired by the update of mean value in Batch Normalization (BN) [18], we update the memory in a moving-average fashion. Specifically, in a training iteration and in the $i^{th}$ feature processing stage, given $H$ samples that are routed to the $j^{th}$ PU, we have their features as $f_i^1, ..., f_i^H$. We update the memory as follows:

$$m_{i,j} := \alpha m_{i,j} + (1 - \alpha) \frac{\sum_{h=1}^{H} \text{GAP}(f_i^h)}{H}, \tag{9}$$

where $\alpha$ is a hyper-parameter and denotes the momentum.

In this step, the computational graph (i.e., data flow) is dynamic due to the routing mechanism. But, the routing-by-memory mechanism uses the non-parametric nearest neighbor algorithm to select the procedures, and the memory is updated in a moving-average fashion. Thus, we can use the original

stochastic gradient descent algorithm with a softmax cross-entropy loss function to train our RMN as the original ResNet-18. We also let each routing operation have a 0.05 probability of doing random routing, which can train procedures more stable.

# 4  Experiments

In this section, we will first introduce our experimental setup. Then we present ablation experiments on our proposed main components and hyper-parameters. Finally, we show our results on some image classification benchmarks.

## 4.1  Experimental Setup

In this section, we first introduce the datasets we used. Then we introduce the network architectures. Finally, we present the training details of our RMN.

### 4.1.1  Datasets

In this paper, we take three image classification benchmarks, Tiny ImageNet [24], ImageNet 2012 [30], and CIFAR-100 [22] to evaluate our method. ImageNet 2012 consists of 1.2M training images and 50,000 validation images for 1000 classes. Tiny ImageNet is a subset of ImageNet. It consists of 200 classes, and each class has 500 training images and 50 validation images. CIFAR-100 consists of 100 classes, and each class has 500 training images and 100 validation images. We use Tiny ImageNet for ablation experiments in Section 4.2. In Section 4.3, we show our results on all three benchmarks.

### 4.1.2  Network Architectures

Our RMN can be applied to most existing CNNs architectures, and we take some widely-used architectures, i.e., VGG, ResNet, and EfficientNet, in our experiments. In the ablation experiments, we take ResNet-18 as the backbone network to train our RMN. In the section of evaluations on all benchmarks, we present the results of using all architectures. For VGG-16, we split the network by pooling layers to different stages and build the PUs. We also use batch normalization for VGG-16. Since the image resolutions of Tiny ImageNet and CIFAR-100 are $64 \times 64$, and $32 \times 32$, respectively, we drop the first two down-sampling operations for ResNet and EfficientNet when training on these two benchmarks. For VGGNet, we drop the first two down-sampling operations on Tiny ImageNet but keep them on CIFAR-100. For the ImageNet, we use the resolution of $224 \times 224$.

### 4.1.3  Training Details

**Learning rate.** We first follow the warmup strategy [10] to increase the learning rate from 1e-5 to 0.48 in the first five epochs. Then we use the cosine learning rate strategy for the rest of the epochs, and we decrease the learning rate to 1e-5 at the final epoch.

**The number of training epochs.** For CIFAR-100 and Tiny ImageNet, the total numbers of training epochs for VGG-16, ResNet-18, ResNet-50, and EfficientNet-B0 are 120, 120, 200, and 300, respectively. For ImageNet, we use 160, 160, 220, and 400 epochs for them, respectively.

**The number of PUs.** In this paper, $N$ denotes the number of PUs per feature processing stage. Increasing $N$ can improve the model capacity but brings in more parameters. Seeing Section 4.2 for the ablation experiments on $N$. We set $N = 8$ for the accuracy and cost trade-off in other experiments. Please also note that, compared with memory consumption on feature extraction, the parameters consume little memory (less than 1% overall consumption when doing inference on ImageNet with batch size 100). Hence, the extra GPU memory consumption by our method is negligible. Besides, extra parameters bring an extra storage cost. But storage is abundant in real-world applications since disks are cheap.

**Batch Size.** We use batch size 256 for all baselines. For our RMN, in each feature processing stage, there are multiple PUs, and different features from the same batch are fed to different PUs. So, the batch size of each PU is much smaller than the total batch size, and accordingly, we have to enlarge the batch size to train our RMN. Simply, we multiply the original batch size (256) by $N/2$ though the

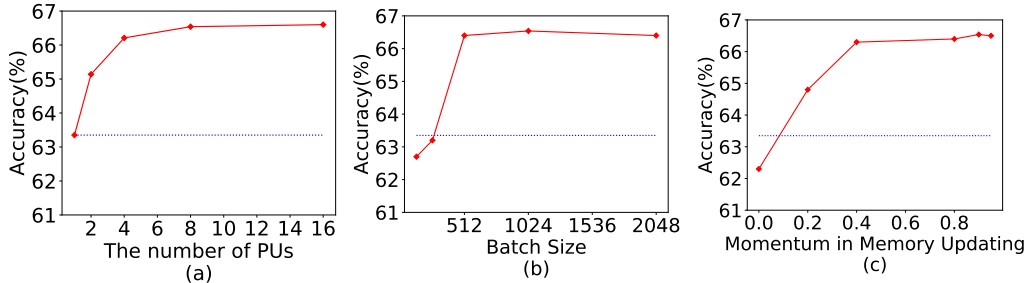

Figure 4: Experimental results on Tiny ImageNet. The blue dashed line denotes the baseline ResNet-18 with CA modules (without conditional routing inputs). (a) More PUs bring higher accuracy, and the accuracy got converged on 8 PUs, so we set it to 8 in other experiments. (b) Since RMN assigns different samples to different PUs, it requires a large batch size to maintain the training stability. We set it to 1024 in other experiments. (c) Higher momentum makes the memory updating smoother and more stable and leads to better results.

numbers of data instances among different PUs are imbalanced, which means that we use $N = 8$ and batch size 1024 in this paper. Please see Section 4.2 for the ablation experiment of batch size.

**Momentum in Memory Updating.** The momentum $\alpha$ is a hyper-parameter used to make the memory updating smooth and stable. However, too high momentum will make the memory out of date, while too small momentum leads the memory updating unstable. We set it to $0.9$ in our paper. Please see Section 4.2 for its ablation experiment.

**Others.** Regarding other training details, we use stochastic gradient descent with momentum 0.9 and weight decay 1e-5 for ResNet and VGGNet. We use the RMSProp optimizer for EfficientNet. We use Synchronized Batch Normalization (SyncBN) supported by the Nvidia APEX library. For CA modules, we use the reduction ratio of 16. For data augmentation, we use random augmentation introduced in the ResNet [13] paper. We use $8\times$V100 (32GB memory version) with PyTorch.

## 4.2 Ablation and Hyper-Parameters Experiments

### 4.2.1 Number of PUs

The key component in our RMN is the PU. So $N$, the number of PUs, is a critical hyper-parameter. Increasing the number of PUs will improve the model's capacity and lead to higher accuracy. From Figure 4 (1), we can see that the accuracy increased along with larger $N$ and got converged when $N = 8$. However, larger $N$ increases the number of parameters and requires a larger batch size (batch size= $256 \times N/2$). We make an accuracy and cost trade-off and set $N$ to 8 in other experiments.

### 4.2.2 Batch Size

Our RMN introduces multiple branches with different PUs in the network, and the examples are assigned to the different branches. So, RMN requires a larger batch size than conventional CNNs. According to the results in Figure 4 (b), too small batch sizes (128 or 256) lead to even lower accuracy than the baseline, which is because the memory updating and routing are unstable. Thus, network training becomes more difficult. Moreover, with the 128 batch size, we found about 40% PUs died after 50 training epochs, which means that these PUs' memories are far from the distribution of the current features, and thus no data can be assigned to these PUs. We consider that the small number of data instances assigned to the PUs will make the memory updating unstable and draw the corresponding memory features to the outliers. Too large batch size (2048) also leads to not good results. Simply, we multiply the original batch size (256) by $N/2$ though the numbers of data instances among different PUs are imbalanced, which means that we set the batch size to 1024 in other experiments.

We also increased the batch size to 1024 for baselines and reported the results for fair comparisons. We find that there is clear overfitting using batch size 1024 for the baselines, especially for ResNet-50 and EfficientNet-B0. These results are lower than the results of our method. Due to the limited pages, please see the results on our supplemental material.

### 4.2.3 Momentum in Memory Updating

We use the moving average to update the memories. Thus the momentum $\alpha$ is a key hyper-parameter. Too small $\alpha$ will lead the memories to change quickly. The higher $\alpha$ is, the more stable and smoother the memory updating becomes, while too large $\alpha$ causes a very slow updating. According to Figure 4 (c), we set $\alpha$ to 0.9 in other experiments.

### 4.2.4 Conditional Attention

The Conditional Attention (CA) module is a SE-like modules to further improve accuracy by introducing a little extra cost. According to the results in Table 1, it improves the ResNet-18's accuracy by 2.2% on Tiny ImageNet.

| Model | CA | Acc. | #Params | #FLOPs |
|---|---|---|---|---|
| RMN (ResNet-18) | | 64.3% | 89.4M | 2.25B |
| RMN (ResNet-18) | ✓ | 66.5% | 89.5M | 2.26B |

Table 1: Ablations of the CA module on Tiny ImageNet.

### 4.3 Evaluation on Benchmarks

We evaluate our methods on three image classification benchmarks: Tiny ImageNet, ImageNet, and CIFAR-100. According to the evaluation results (Table 2) on Tiny ImageNet, our RMN significantly improves the accuracy while not increasing the computational cost. Besides, the improvements for VGGNet and ResNet are much more significant than EfficientNet. It is because EfficientNet's architecture is already well-designed, compact, and contains SE modules. Thus, our method may bring some redundancies, but it still improves the accuracy by

| Model | Acc. | #Params | #FLOPs |
|---|---|---|---|
| VGG-16 [32] | 63.16% | 135.0M | 14.10B |
| ResNet-18 [13] | 61.78% | 11.3M | 2.25B |
| ResNet-50 [13] | 67.28% | 23.9M | 5.25B |
| EfficientNet-B0 [34] | 67.14% | 4.3M | 0.47B |
| RMN (VGG-16) | 66.57% | 237.8M | 14.11B |
| RMN (ResNet-18) | 66.53% | 89.5M | 2.26B |
| RMN (ResNet-50) | 69.93% | 189.1M | 5.26B |
| RMN (EfficientNet-B0) | 68.85% | 32.5M | 0.47B |

Table 2: Evaluation on Tiny-ImageNet. Ten runs average results (skip when occurring PU death).

1.8%. Seeing from the results (Table 3) on the more challenging benchmark, ImageNet, our method still performs well. Moreover, using our RMN, ResNet-50 can outperform EfficientNet-B0. For the CIFAR-100 benchmark, although the image resolution is small ($32 \times 32$) our method can still gain considerable improvements. Moreover, ResNet-18-based RMN achieves impressive accuracy and even outperforms the original ResNet-50.

| Model | Acc. | #Params | #FLOPs |
|---|---|---|---|
| VGG-16 [32] | 72.92% | 138.3M | 15.51B |
| ResNet-18 [13] | 70.72% | 11.7M | 1.81B |
| ResNet-50 [13] | 76.08% | 25.6M | 4.11B |
| EfficientNet-B0 [34] | 76.46% | 5.3M | 0.39B |
| RMN (VGG-16) | 76.16% | 241.1M | 15.52B |
| RMN (ResNet-18) | 73.74% | 89.9M | 1.82B |
| RMN (ResNet-50) | 78.31% | 190.7M | 4.12B |
| RMN (EfficientNet-B0) | 78.08% | 33.5M | 0.40B |

Table 3: Evaluation on ImageNet. Three runs average results (skip when occurring PU death).

| Model | Acc. | #Params | #FLOPs |
|---|---|---|---|
| VGG-16 [32] | 73.16% | 33.8M | 0.43B |
| ResNet-18[13] | 75.62% | 11.2M | 0.56B |
| ResNet-50[13] | 77.53% | 23.7M | 1.31B |
| EfficientNet-B0[34] | 78.04% | 4.2M | 0.12B |
| RMN (VGG-16) | 75.74% | 136.8M | 0.45B |
| RMN (ResNet-18) | 77.94% | 88.4M | 0.57B |
| RMN (ResNet-50) | 79.06% | 184.6M | 1.32B |
| RMN (EfficientNet-B0) | 78.88% | 32.4M | 0.12B |

Table 4: Evaluation on CIFAR-100. Ten runs average results (skip when occurring PU death).

### 4.4 Comparisons with Related Work

RNR [28] and DeepMOE [38] are the two most related works. Our motivations and methods are different, and we discuss and make comparisons in this subsection.

RNR learns an RNN-based routing network by reinforcement learning to dynamically select experts. It trains the routing network and the backbone network alternately. DeepMOE learns a CNN-based decision network to activate specific layers in the backbone network dynamically. Unlike these methods, our RMN solves routing in a new perspective, routing by memory, rather than introducing complicated routing networks. It is more straightforward and elegant. Besides, since the routing is based on non-parametric nearest memory search, RMN is lightweight, easy to train, and requires no modification in the back-propagation rule. Here we make the experimental comparisons with RNR.

For RNR, we follow its codes to align our baselines' training details and then retrain our RMN on CIFAR-100. Based on ResNet-18 backbone, our RMN ($N = 4$, without CA) achieves 78.96%

accuracy while costs 14.2ms per image (using Xeon 8255C CPU), which is superior to RNR's (#branch=4) results-78.42% accuracy with 59.8ms.

For DeepMOE, we make comparisons on respective baselines. DeepMOE improves ResNet-50's accuracy on ImageNet from 76.15% to 77.12%, while our RMN improves ResNet-50 from 76.08% to 77.78% ($N = 8$, without CA). Besides, our RMN is more efficient since it does not introduce extra decision networks.

## 5 Conclusion

In this paper, we proposed a specific mechanism, routing by memory, for conventional feed-forward networks. We integrated it with the existing CNN architectures and built the Routing by Memory Network (RMN). Specifically, it introduces the Procedural Unit (PU) to the CNNs, which consists of a memory (a representative feature) with a procedure (some convolutional blocks). We employed memories to forward different features to their expert PUs. Networks with the proposed mechanism can be trained efficiently using a four-step training strategy. According to the results on Tiny ImageNet, ImageNet, and CIFAR-100, our RMN significantly improves VGG-16, ResNet-18, ResNet-50, and EfficientNet-B0 while not increasing the computational cost.

## 6 Limitation Analysis

The main limitations of our method are extra parameters, larger training batch sizes, unstable training, and parallel inference efficiency.

First, since our method clones the experts multiple times, it introduces extra parameters and consumes additional memory and storage. But, as we discuss in Section 4.1.3, the model parameters consume little memory. Besides, in many real-world scenarios, the model storage cost is negligible. In other words, our method can trade redundant memory and storage for accuracy improvement.

Second, since our network is multi-branch, each branch (expert) requires enough training samples in each training iteration. Thus, our method requires a larger training batch size ($4\times$ for 8-way RMN). But V100 (32GB) is enough to train most existing CNN models. Besides, some techniques can use larger batch sizes in limited memory, such as gradient accumulation, memory-saving CNN training framework (e.g., MXNet), and dynamic memory allocation.

Third, the model training is unstable since some PUs may die, especially when using not good training hyper-parameters. If some PUs die, the accuracy will usually be lower than that of the baseline.

Finally, since the data flow in our method is multi-path, its parallel inference efficiency is lower than that of the baseline when inference batch size is greater than one.

Last but not least, our rule of assigning similar features to the same expert is a mixed blessing. It is explainable, non-parametric, and fast. But compared with parametric routing, our rule may impede the model itself from achieving better results.

## Acknowledgments and Disclosure of Funding

This work is supported by JST AIP Acceleration Research Grant Number JPMJCR20U1 and JSPS KAKENHI Grant Number JP20H04205, Japan. This work is also supported by Japan JSPS Research Fellowship for Young Scientists (DC) with JSPS KAKENHI Grant Number JP21J13152.

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
