# Neural Routing by Memory
## - Supplementary Material -

**Kaipeng Zhang**[1,2]  **Zhenqiang Li**[1]  **Zhifeng Li**[2*]  **Wei Liu**[2*]  **Yoichi Sato**[1*]

[1]Institute of Industrial Science, The University of Tokyo
[2]Tencent Data Platform

{kpzhang, lzq}@iis.u-tokyo.ac.jp    michaelzfli@tencent.com
wl2223@columbia.edu    ysato@iis.u-tokyo.ac.jp

## 1   Runtime

In the main paper, we report the theoretical computation cost, i.e., the number of FLOPs. Here we report the actual runtime on ImageNet ($224 \times 224$) using Xeon 8255C CPU. ResNet-18, ResNet-50, and EfficientNet-b0 cost 25.1ms, 51.6ms, and 30.2ms per image, respectively. Corresponding RMNs($N = 8$), without CA: 26.4ms, 54.2ms, 33.0ms; with CA: 27.1ms, 55.2ms, 33.6ms. These results are average runtime on the whole ImageNet validation set. They demonstrate that our method is also efficient in real inference.

## 2   Batch Size

As we introduced in our main paper, we use batch size 256 for baselines and increase the batch size to 1024 for our RMN. Here we increase the batch size to 1024 for baselines and report the results. From the results on Table 1, it is clear that using batch size 1024 for the baselines leads to accuracies drop and performs lower results than the results of our method.

| Benchmark | Method | Batch Size | VGG-16 | ResNet-18 | ResNet-50 | EfficientNet-B0 |
|---|---|---|---|---|---|---|
| CIFAR-100 | Original | 256 | 73.16 | 75.62 | 77.53 | 78.04 |
| | Original | 1024 | 70.55 | 73.36 | 73.02 | 73.84 |
| | RMN | 1024 | 75.74 | 77.94 | 79.06 | 78.88 |
| Tiny ImageNet | Original | 256 | 63.16 | 61.78 | 67.28 | 67.14 |
| | Original | 1024 | 61.71 | 60.15 | 65.34 | 65.82 |
| | RMN | 1024 | 66.57 | 66.53 | 69.93 | 68.85 |
| ImageNet | Original | 256 | 72.92 | 70.72 | 76.08 | 76.46 |
| | Original | 1024 | 72.76 | 70.82 | 76.02 | 76.53 |
| | RMN | 1024 | 76.16 | 73.74 | 78.31 | 78.08 |

Table 1: Results of using different batch sizes.

## 3   Memory Analysis

We train a ResNet-18-based RMN with $N = 4$ on Tiny ImageNet to analyze the memory. After testing, we show each memory's top K nearest validation images in Figure 1. According to the results, memories mainly focus on low-level features in the first two stages, such as color, texture, and object shape. For example, $m_{1,1}$ mainly focuses on yellow color and $m_{1,4}$ mainly catches mixed colors

---

*    Co-corresponding authors

35th Conference on Neural Information Processing Systems (NeurIPS 2021).

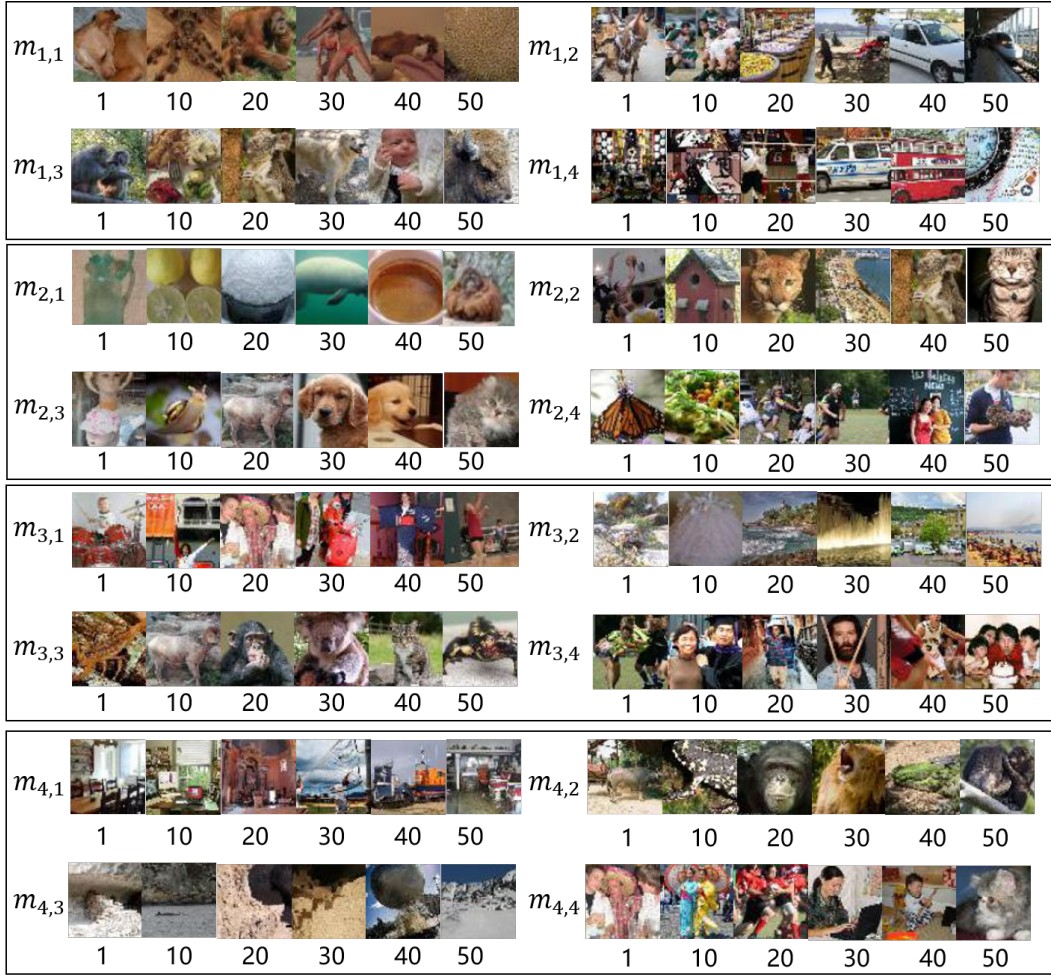

Figure 1: Memory analysis. We train ResNet-18-based RMN ($N = 4$) on Tiny ImageNet. We show the validation images that are $K^{th}$ (the numbers under the images) closest to different memories. From the results, Memories in the first two stages mainly focus on low-level features, while memories in the last two stages considers semantic information in the last two stages.

while $m_{2,1}$ mainly takes the circles. In the last two stages, memories can capture more semantic information. For example, $m_{4,2}$ and $m_{4,4}$ focus on animals and human activity, respectively.