# OpenReview forum: "Neural Routing by Memory"
_NeurIPS.cc/2021/Conference — NeurIPS 2021 Poster_

### Official Review · Reviewer_fdXd · 2021-07-10

**Rating:** 4
**Confidence:** 4

**Summary:**

The paper proposes to use extra network blocks (called memory units by the authors) to route different features in a network to different conv layers based on the k-nearest neighbors computation. Additional attention layers are also used to further scale the routing results as the input for the next layer. The proposed routing approach is applied to VGGNet, ResNet, and EfficientNet and shows accuracy improvement but with significant increases in memory/parameter cost.

**Limitations And Societal Impact:**

I appreciate the authors discussing the limitation in terms of extra parameters used in the proposed routing approach. However, without addressing this issue, it is hard to justify why the proposed routing method is more preferred compared to the recent architecture search methods that performs branch selection and then re-training.

**Main Review:**

Pros:
* Thorough literature survey in the field of Mixture of Expert and Routing.
* Figure 1 & 2 are helpful in comprehending the proposed modification to conventional routing architecture.
* Detailed hyperparameter configurations are provided for the reproducing purpose.

Cons:
* The major change to the conventional routing-based architectures in the proposed method is to replace the Gumbel Softmax routing with the k-nearest neighbors routing. However the authors only provide experimental results w/ and w/o the proposed routing method on three different backbone models. It is critical to understand why non-parametric routing by k-nearest neighbors is more preferable than the Gumbel Softmax routing. The comparison with RNR [2] and DeepMOE [3] in the appendix are not based on Gumbel Softmax routing.
* As mentioned in the limitation section, the proposed routing algorithm requires significantly more parameters compared to the corresponding backbone networks. Specifically, on ImageNet, the proposed RMN requires 7.6 times the number of parameters (ResNet-18 from 11.7M increases to 89.9M parameters) in the resulting network architecture. The number of parameters in the proposed routing-based ResNet-18 is already more than 4 times the number of parameters in ResNet-1202. To me this does not seem to be an effective usage of a network’s capacity.
* The routing concept is essentially doing differentiable architecture search but without the final path selection and final re-training. How is the proposed routing compared to recent DART [A1] based approaches? Do we really need to keep all the branches in the routing block? If not, can we simply perform branch selection and re-training after the selection is done?

[A1] DARTS: Differentiable Architecture Search. ICLR 2019.


**Time Spent Reviewing:**

8

---

> ### Author Response · Authors · 2021-08-24
> **Response to Reviewer fdXd**
>
> We thank you for your careful evaluation and suggestions on improving our work. We address your main concerns below.
>
> **Q1. The major change to the conventional routing-based architectures in the proposed method is to replace the Gumbel Softmax routing with the k-nearest neighbors routing. However, the authors only provide experimental results w/ and w/o the proposed routing method on three different backbone models. It is critical to understand why non-parametric routing by k-nearest neighbors is more preferable than the Gumbel Softmax routing. The comparison with RNR [2] and DeepMOE [3] in the appendix are not based on Gumbel Softmax routing.**
>
> A1. Thank you for raising the Gumbel Softmax, which is one of the ways to train decision networks. Yes, RNR and DeepMoE do not use Gumbel Softmax to train decision networks. RNR uses reinforcement learning to train an RNN-based decision network, while DeepMOE uses continuous weighting with a two-step training algorithm to train a CNN-based decision network. Since we perform non-parametric routing, we do not need Gumbel Softmax, reinforcement learning, or other tricks to train decision networks. Gumbel Softmax is more often used in MoE models for NLP tasks and generative models and neural architecture search for CV tasks. Since we mainly surveyed and discussed MoE models for CV, we missed Gumbel Softmax. In our future research, we will try Gumbel Softmax if it works well on MoE models for CV tasks.
>
> **Q2. As mentioned in the limitation section, the proposed routing algorithm requires significantly more parameters compared to the corresponding backbone networks. Specifically, on ImageNet, the proposed RMN requires 7.6 times the number of parameters (ResNet-18 from 11.7M increases to 89.9M parameters) in the resulting network architecture. The number of parameters in the proposed routing-based ResNet-18 is already more than 4 times the number of parameters in ResNet-1202. To me this does not seem to be an effective usage of a network's capacity.**
>
> A2. Thank you for your comment. We agree that there may be significant redundancy in our network and that introducing extra parameters may be the main limitation of our method. Reducing the number of parameters is one of the research lines of our future works. But we want to claim that our model size is still small for current GPUs (e.g., 32GB RAM for V100 or RTX3090). And compared with memory consumption on feature extraction, the parameters only consume little memory, especially in batch inference.
>
> **Q3. The routing concept is essentially doing differentiable architecture search but without the final path selection and final retraining. How is the proposed routing compared to recent DART [A1] based approaches? Do we really need to keep all the branches in the routing block? If not, can we simply perform branch selection and retraining after the selection is done?**
>
> A3. Thank you for pointing out the way of differentiable neural architecture search. It is another research topic but related to our work. We think it is like training and selecting the optimal experts, while MoE is like training the multi-diverse experts. Both of them are worth further study.

---

> > ### Comment · Reviewer_fdXd · 2021-08-27
> > **Final rating**
> >
> > Thanks to the authors for preparing the answers. However I did not find the answers directly addressing my concerns. I will keep my original rating.

---

### Official Review · Reviewer_bRNA · 2021-07-15

**Rating:** 6
**Confidence:** 4

**Summary:**

This paper proposes a routing mechanism to dynamically adjust the DNN forwarding path conditioned on the input. This relieves the learning difficulty of a model by processing different features with specialized procedures. A four-step training strategy is introduced to support this dynamic routing process. Experiments on various backbone networks validate the effectiveness of the proposed method.

**Main Review:**

Originality:
The idea of enforcing divide-and-conquer in DNN forwarding based on feature grouping looks new to me, although the internal technical components (e.g. k-means clustering, moving average, squeeze-and-excitation, etc) are commonly used tools.

Quality:
The proposed method is technically sound. It is intuitive that networks specialized to handle certain groups of features can relieve the learning complexity and potentially improve performance.
My concern comes from the validation of the feature grouping. The visualization of the memory analysis has been put in the supplementary material. The separation of different features in different PUs (Fig. 1 in supplementary) is not obvious to me. Can you show more visualizations on more architectures and datasets to validate this specialized feature handling claim?
I also wonder about the statistics of the usage of the routing dynamics, e.g. how many times on average the branch-switch happens across a dataset? what is the frequency of branch-switch at different stages in a model? any comparison between forcing the usage of different branches?

Clarity:
The paper is clear and easy to follow.

Significance:
I think this dynamic architecture routing idea is beneficial to this community.

**Time Spent Reviewing:**

3

---

> ### Author Response · Authors · 2021-08-24
> **Response to Reviewer bRNA**
>
> We thank you for your careful evaluation and suggestions on improving our work. We address your main concerns below.
>
> **Q1. Can you show more visualizations on more architectures and datasets to validate this specialized feature handling claim?**
>
> A1. Thank you for your suggestion. We cannot include figures or links in the response, but we will add more visualizations to the revised paper. Besides directly showing the routing results as we did in supplementary materials, we also tried to generate different inputs using different memories as objectives, resembling the deep dream method (https://github.com/google/deepdream) but in a supervised gradient descent fashion. We find that different memories reflect different semantic patterns as we expected. We will also add this visualization method to the revised paper.
>
> **Q2. I also wonder about the statistics of the usage of the routing dynamics, e.g., how many times on average the branch-switch happens across a dataset? What is the frequency of branch-switch at different stages in a model?**
>
> A2. Thank you for your suggestion. If the mentioned branch-switch means that a sample chooses a different branch number in later stages (choose the first procedure in the first stage and choose the second procedure in the second stage), it means nothing since the branch number means nothing. It is because branches in different stages are specialized to extract different features (e.g., the first branch in the first stage and the first branch in the second stage are specialized to extract different features).
>
> If the mentioned branch-switch means that a training sample chooses different procedures (within a stage) along with the training process, it frequently happens in around the first 20% epochs after the memory initialization on ImageNet training.
>
> We will show statistics about the routing results with detailed charts in our revised paper to analyze our method in-depth.
>
> **Q3. Any comparison between forcing the usage of different branches?**
>
> A3. We multiply the number of channels in each stage by 8 (i.e., using all branches) to retrain baselines, which should be an upper bound of our method. Since deep models are too big to train (high memory consumption and computational cost), we only train ResNet-18 on CIFAR-100. ResNet-18 achieves 79.62% accuracy, while our RMN and the original baseline achieve 77.9% and 75.5% accuracies, respectively.

---

### Official Review · Reviewer_cGyg · 2021-07-16

**Rating:** 3
**Confidence:** 4

**Summary:**

This paper proposes a conditional computation model based on memory (not sure this is an appropriate term; see below). The routing-by-memory mechanism saves representative features for each procedure and later compares them with input features to select the most appropriate procedure. To stabilize routing-by-memory networks, a four-step training strategy is also introduced. The experiments show that routing-by-memory networks outperform their non-RMN baselines on image classification tasks with the similar number of flops.

**Limitations And Societal Impact:**

.

**Main Review:**

My main concern is the novelty. The deep models with conditional computation have been around for years, and using representative features for routing is not new either [1, 2]. I think the only possible novelty here is adjusting representative features during the training. But it is unclear if this way is better than fixing routes with a pre-trained feature extractor in the beginning because the authors did not compare RMNs to any conditional computation baselines.

The fact that accuracy improved only a little while the number of parameters increased by several folds is disappointing ([3] observed a similar pattern in their work). It is not practical in terms of computational cost, because if the memory requirement becomes larger than a single GPU/TPU capability, the communication overhead will be involved.

Also, I am not sure if using the term memory is correct here. In this paper, the memory content is modified between training steps. I think the memory commonly refers to a "scratchpad" that can save temporary information during a single inference pass.

[1] Park, David Keetae, et al. "Megan: Mixture of experts of generative adversarial networks for multimodal image generation." arXiv preprint arXiv:1805.02481 (2018).

[2] Mullapudi, Ravi Teja, et al. "Hydranets: Specialized dynamic architectures for efficient inference." Proceedings of the IEEE Conference on Computer Vision and Pattern Recognition. 2018.

[3] Ramachandran, Prajit, and Quoc V. Le. "Diversity and depth in per-example routing models." International Conference on Learning Representations. 2018.

**Time Spent Reviewing:**

3

---

> ### Author Response · Authors · 2021-08-24
> **Response to Reviewer cGyg**
>
> We thank you for your careful evaluation of our paper. We bring a novel research perspective to routing and simplify the routing process by a non-parametric routing-by-memory mechanism, which is more explainable, quick, and easy to train. We address your main concerns below.
>
> **Q1. The deep models with conditional computation have been around for years, and using representative features for routing is not new either [1, 2]. I think the only possible novelty here is adjusting representative features during the training. But it is unclear if this way is better than fixing routes with a pre-trained feature extractor in the beginning because the authors did not compare RMNs to any conditional computation baselines..**
>
> A1. Thank you for raising the related papers [1,2]. But they do not use representative features (i.e., memory in our paper) for routing, and their frameworks differ significantly from ours.
>
> First, the paper [1] does not use representative features. It is a GAN framework and uses features from all generators to choose a generator by training an extra gating network.  Besides, compared to typical MoE methods (including ours), it needs to activate all paths (i.e., all generators) before selecting a path, which introduces a heavy computational burden.
>
> Second, the paper [2] does not use representative features, either. It constructs super-classes (i.e., partitioning the original 1,000 classes) by clustering the features extracted from a pre-trained model. Different network branches are specialized to account for different super-classes. In addition, it introduces an extra subnetwork to make super-class predictions to choose top K branches in a multi-branch network.
>
> We want to claim that besides the memory, we propose a non-parametric routing-by-memory mechanism that is more explainable, quick, and easy to train.
>
> Also, please refer to our supplementary materials, where the experiments demonstrate that our method outperforms the two most related MoE-based methods, RNR (using RNNs for routing) and DeepMoE (using CNNs for routing).
>
> **Q2. The fact that accuracy improved only a little while the number of parameters increased by several folds is disappointing ([3] observed a similar pattern in their work). It is not practical in terms of computational cost, because if the memory requirement becomes larger than a single GPU/TPU capability, the communication overhead will be involved.**
>
> A2. Yes, as discussed in our limitation analysis at the end of our paper, introducing extra parameters may be the main limitation of our proposed method. Reducing the number of parameters is one of the research lines of our future works. But we want to claim that our model size is still small for current GPUs (e.g., 32GB RAM for V100 or RTX3090). And compared with memory consumption on feature extraction, the parameters only consume little memory, especially in batch inference.

---

> > ### Comment · Reviewer_cGyg · 2021-08-27
> > **After rebuttal**
> >
> > I thank the authors for more explanations. However I am not still convinced. Therefore my score remains unchanged.

---

### Official Review · Reviewer_Qanu · 2021-07-17

**Rating:** 4
**Confidence:** 5

**Summary:**

This work proposes a nonparametric method for routing in convolutional network image classifiers, where routing is the selection of a subset of modules to execute on a given input. Routing by memory associates each module with a feature, the "memory", that is scored against the input feature to select the module with the nearest memory. The routing is done at each stage of the network, and selects a single memory and its associated module, so the resulting architecture is always a chain from input to output (and not more general DAG). The memories are updated by the mean of the features assigned to it as a moving average during training. The combination of nearest neighbor matching and momentum updates yields a routing rule with few parameters, as it has only those of the embedding for memories and an optional squeeze-and-excitation-like adaptor for each module. This approach does not require reinforcement learning or stochastic optimization tricks like the gumbel-softmax to learn the representation or routing. This routing scheme requires a phased training approach: 1. learn a feature representation with a standard network, 2. widen the network by duplicating the modules in each stage, 3. cluster features to initialize a memory for each module, 4. continue updating the modules by gradient descent and the memories by clustering and momentum. In total, this routing scheme increases the number of parameters without increasing inference computation time. Experiments on standard architectures, specifically VGG-16, ResNet-{18,50}, and EfficientNet-B0, show higher accuracy with the proposed routing than without. Accuracies improve by 1.5-4 points with 2x-8x the parameter count. Ablations examine the methods parts and hyperparameters on the minor Tiny ImageNet dataset: both the memory routing and squeeze-and-excitation-like adaptor help and the number of parallel modules for routing, batch size, and momentum for memory updates are all effective hyperparameters. No experiments compare against routing baselines.

**Ethical Concerns:**

None.

**Limitations And Societal Impact:**

The limitations of increased parameter counts and complicating training are addressed satisfactorily in the conclusion. This work has no particular potential negative social impacts, only having the general possible impacts of an alternative deep network architecture for visual recognition, and so no special comment is needed.

**Main Review:**

Strengths
- The baseline architectures are diverse: VGG, ResNets, and EfficientNet(-B0).
- The choices of number of modules (PUs), batch size, and momentum over the nonparametric parts (memories) are ablated.
- The proposed method is essentially "free" at inference time in terms of FLOPs, since only one module is routed at each depth. (However, there is no GPU runtime profiling. Often FLOPs reduction does not convert to time reduction.)

Weaknesses
- No competing method is compared against. Instead, all results compare standard baseline networks with and without the proposed neural routing by memory. Comparisons are promised in the supplement, but only one of many methods is reproduced for fair comparison, and the accuracy on this single experiment differs by only 0.5 point.
- Routing is restricted to a single choice of procedure, so routing always results in a "line" architecture that cannot run more than one module in each step. Other work, such as CondConv, has "soft" routing that can make use of multiple modules at the same depth.
- The training is more complicated, in that it has to be divided into four stages. Other routing methods, like MoE methods, can simply be trained in one go.
- Claims are not justified. "Routing is based on non-parametric nearest memory" does not imply the method is "easy to train" or prove that alternatives require "complicated decision networks" CondConv [36] for example is quite simple, and does not require multi-stage training. For the ease of training, it is unclear a priori if routing by nearest neighbor will be easier or harder to train, since the nearest neighbor could switch from one iteration to the next, which might radically change gradients. This claim needs an experiment that measures training, by for instance examining the entropy of routing decisions and the task accuracy across epochs.
- Experiments are not controlled. The accuracy improvement is potentially confounded by (1) change in batch size and (2) momentum updates to the memories in the RMN. Batch size is well-known to be a sensitive hyperparameter, and momentum updates/weight averaging are both the subject of research for improved optimization (see Polyak averaging, weight averaging as in "Averaging Weights Leads to Wider Optima and Better Generalization" UAI'18, etc.). Although larger batch sizes rarely help in this regime, it nevertheless needs confirmation for controlled experimentation.

Related Work: Dynamic/predicted parameter methods [4, 5, 36] are discussed but not compared against. Why not? They share the goal of increasing parameter counts to thereby increase accuracy while controlling the amount of computation for inference. Note there are more such methods in existence, like Dynamic Filter Nets [NeurIPS'16] and Involution [CVPR'21, after submission deadline].

Clarity: The vocabulary of the paper is unique to this work, and so presents an unnecessary barrier to comprehension. To give a few examples: "procedure", "stem network", "conditional" attention. Is a procedure in any way different than the more common usage of the word "block"?

Decision: The contribution cannot be gauged as a research or engineering contribution  without comparison to related work. The experiment design throughout this work denies comparison by only evaluating totally standard baseline networks and the proposed routing scheme. That is, no alternative routing scheme is tried. Nor are other interventions increasing parameter count and/or batch size tried as controls for two main effects of the proposed routing scheme. These empirical limitations along with missing related work drive the decision for rejection.

For Rebuttal:
- Please compare results with MoE and related methods, such as CondConv, in order to have relevant routing-based comparisons.
- Please report results of control experiments on batch size and weight averaging/momentum on the parameters of standard networks.

**Time Spent Reviewing:**

2.5

---

> ### Author Response · Authors · 2021-08-24
> **Response to Reviewer Qanu**
>
> We thank you for your careful evaluation and suggestions on improving our work. We address your main concerns below.
>
> **Q1. Please compare results with MoE and related methods, such as CondConv, in order to have relevant routing-based comparisons.**
>
> A1. Thank you for your suggestion. The relationships between our method and other MoE methods are explained in the related works section (L102-L123). Our supplementary materials have demonstrated that our proposed routing-by-memory mechanism outperforms the two most related MoE-based methods, RNR (using RNNs for routing) and DeepMoE (using CNNs for routing), in terms of both accuracy and runtime efficiency. We would like to emphasize that, beyond the accuracy, our work brings a novel research perspective to routing and simplify the routing process by a non-parametric routing-by-memory mechanism. Unlike introducing parametric routing modules, our proposed mechanism is more explainable, quick, and easy to train.
>
> Regarding the comparison with CondConv, based on ResNet-50, CondConv improves the accuracy from 77.7% to 78.6% (+0.9%) while our method improves the accuracy from 76.1% to 78.3% (+2.2%). Based on EfficientNet-B0, CondConv improves the accuracy from 77.2% to 78.3% (+1.1%), while our mechanism improves the accuracy from 76.5% to 78.1% (+1.6%). We can see that our method shows better improvements in accuracy than CondConv. Besides, our method performs block-wise routing, whereas CondConv performs layer-wise parameters combination. Thus, our routing mechanism is compatible with CondConv.
>
> **Q2. Please report the results of control experiments on batch size and weight averaging/momentum on the parameters of standard networks.**
>
> A2. Thank you for your suggestion. We have tuned the hyper-parameters (e.g., learning rates, batch sizes, and the number of epochs) to make standard networks perform well before applying our proposed RMN.
>
> Regarding batch size, similar to our experiments in Section 4.2.2, we evaluate the standard (baseline) ResNet-18 on Tiny-ImageNet using different batch sizes. It achieves the accuracy of 61.5%, 61.6%, 61.2%, 60.9%, and 60.2% using the batch size of 128, 256, 512, 1024, and 2048, respectively. In Section 4.2.2, our method achieves 62.7%, 63.2%, 66.4%, 66.5%, and 66.4% accuracies, respectively. These results demonstrate that the gain does not originate from increased batch size.
>
> Regarding momentum on the parameters, there may be a misunderstanding. The 'momentum' mentioned in our paper is the momentum in memory updating. We do not change the momentum value in the optimizer.

---

> > ### Comment · Reviewer_Qanu · 2021-09-10
> > **Thank you for the response.**
> >
> > > Our supplementary materials have demonstrated that our proposed routing-by-memory mechanism outperforms the two most related MoE-based methods, RNR (using RNNs for routing) and DeepMoE (using CNNs for routing)
> >
> > As mentioned in the review, it is not appropriate to have the only comparisons with alternative routing methods in the supplement, and for these comparisons to be more limited than with non-routing baselines (the standard architectures in the main tables of the paper). The results in accuracy are quite close (to within 0.5 point) and only one method is reproduced by running the official code.
> >
> > As a sidenote, it is strange to compare the computational efficiency of RNR and the proposed method on CPU. For clock time measurements, it makes sense to measure on GPU, since virtually all deep learning computation is executed on GPUs or other accelerators and not CPUs.
> >
> > > Regarding the comparison with CondConv
> >
> > Thank you for the comparison with CondConv. These results should be added to the main paper for resubmission. It is encouraging that the proposed method can improve as much or a bit more than CondConv in some experiments.
> >
> > As noted by the authors, CondConv is layer-wise while the proposed method is block-wise, so an analysis of a block-wise analysis of CondConv would be informative to determine if there is improvement from this particular aspect of routing.
> >
> > > We would like to emphasize that, beyond the accuracy, our work brings a novel research perspective to routing and simplify the routing process by a non-parametric routing-by-memory mechanism.
> >
> > To reiterate, the claims of simplicity or making the network "easy to train" are not evidenced by results. Please see the "Claims are not justified." point in the review for suggested analyses to ground these claims in experiments.
> >
> > > Regarding momentum on the parameters, there may be a misunderstanding. The 'momentum' mentioned in our paper is the momentum in memory updating.
> >
> > Just to confirm, I did follow this detail, but raised it as a potentially sensitive hyperparameter because it determines the rate at which the "memories" change. The experiments do show the memory momentum is an effective hyperparameter in its own right; my comment was about how the proposed hybrid training by gradients + memory momentum might compare to standard optimization. That is, how large are the norms of the updates across iterations, for instance, to analyze how much the network is changing during training.
> >
> > **Summary**: The positive result with respect to CondConv and the double-checking of batch size effects are improvements. However, more comparison with existing work and more analysis to justify claims is still needed, as mentioned in the initial review. As submitted the main results only compare to non-routing baselines, which is not appropriate or informative when so much work has been done on routing. I have raised my score to 4, to account for the response, but still vote for rejection in this round.
> >
> > The authors are encouraged to more thoroughly compare and analyze their routing method for resubmission.

---

### Decision · Program_Chairs · 2021-09-28

**Decision:**

Accept (Poster)

**Comment:**

This paper has received 4 expert reviews, with a single reviewer providing a slightly favorite rating (but a less informative review),
and 3 extremely critical and informative reviews. The reviewers raised the following points:

= Unconvincing evaluations, as comparisons are done with own baselines only, i.e. without the contributions of the paper, and no comparison with a competing method is in the main paper. One comparison was discussed in the discussion phase, after reception of the author's response, but it was found to be unconvincing. As a summary, the results in accuracy are quite close and only one method is reproduced by running the official code.

= The advantages of the method are unclear wrt to the state of the art

= Experiments are not controlled.

The author's response has been found to be handwavy in many respects, iterating claims of the paper instead of attempting to answer the issues raised by reviewers.

The AC judges that the paper is not yet ready.

**Consistency Experiment:**

NeurIPS has a long history of experimentation. In 2014, NeurIPS ran an experiment in which 10% of submissions were reviewed by two independent committees to quantify the randomness in the review process. This year, we repeated a variant of this experiment to see how the quality of the review process has changed over time.  This paper was part of the experiment and was therefore assigned to two committees (consisting of reviewers, an Area Chair, and a Senior Area Chair) that reached independent decisions.  If both committees made the same recommendation, this recommendation was followed. If a single committee recommended acceptance, the paper was accepted (with the exception of a few cases in which the other committee identified what we considered a fatal flaw, e.g., an error in a key result).

This copy’s committee reached the following decision: **Reject**

The other committee assigned to the paper recommended **Accept (Poster)**.  You can find the other set of reviews, along with any follow up discussion with the authors here:
https://openreview.net/forum?id=rHNF8Kq3u2P